# Long-Term Stable Online Acetylene Detection by a CEAS System with Suppression of Cavity Length Drift

**DOI:** 10.3390/s19030508

**Published:** 2019-01-26

**Authors:** Qixin He, Qibo Feng, Jiakun Li

**Affiliations:** MoE Key Lab of Luminescence and Optical Information, Beijing Jiaotong University, Beijing 100044, China; qbfeng@bjtu.edu.cn (Q.F.); jkli@bjtu.edu.cn (J.L.)

**Keywords:** infrared absorption spectroscopy, cavity-enhanced absorption spectroscopy, gas sensor

## Abstract

A trace acetylene (C_2_H_2_) detection system was demonstrated using the cavity-enhanced absorption spectroscopy (CEAS) technique and a near-infrared distributed feedback (NIR-DFB) laser. A Fabry–Perot (F–P) cavity with an effective optical path length of 49.7 m was sealed and employed as a gas absorption cell. Co-axis cavity alignment geometry was adopted to acquire a larger transmitted light intensity and a higher sensitivity compared with off-axis geometry. The laser frequency was locked to the cavity fundamental mode (TEM_00_ mode) by using the Pound–Drever–Hall (PDH) technique continuously. By introducing a cavity length-locking loop, the drift of the cavity length was suppressed, and the stability of the system was enhanced. To demonstrate the efficacy of the system, a C_2_H_2_ absorption spectrum near 6534.36 cm^−1^ was acquired by tuning the laser operation temperature. Measurements of C_2_H_2_ samples with different concentrations were carried out, and a good linear relationship between C_2_H_2_ concentration and the cavity-transmitted signal voltage was observed. The measurement results showed the system could work stably for more than 2 h without major fluctuations. The Allan variance analysis results demonstrated a detection limit of 9 parts-per-billion (ppb) with an averaging time of 11 s corresponding to a minimum detectable absorption coefficient of 1.1 × 10^−8^ cm^−1^.

## 1. Introduction

Acetylene (C_2_H_2_) plays an important role in illumination, metal welding, and industrial production. The detection of trace C_2_H_2_ has attracted much attention due to its inflammable and explosive characteristics [1,2,3]. Furthermore, C_2_H_2_ is a kind of indicating gas dissolved in transformer oil. The monitoring of C_2_H_2_ concentration is of great significance for working state and performance evaluation of the transformer [4,5]. Among commonly used detection techniques for trace C_2_H_2_, resonant cavity-based infrared laser absorption spectroscopy techniques have attracted a lot of interest with the advantages of high detection sensitivity, fast response, and small sample volume. Techniques such as cavity ringdown spectroscopy (CRDS) [6,7,8] and cavity-enhanced absorption spectroscopy (CEAS) [9,10] make use of high-finesse optical cavities that greatly increase the effective optical path length, thereby improving the detection sensitivity.

CRDS was proposed by Deacon and O’Keefe in 1988 [11]. In CRDS, the decay time of a light pulse trapped in the cavity is measured. By comparing the decay time of the empty cavity with that of the cavity filling with measured gas, the absorption coefficient can be obtained. Since the ring-down procedure occurrs in a fairly short period, CRDS highly depends on fast-response detection electronics, which limits its application [12,13]. Besides, the spectral resolution of CRDS is relatively low due to the use of pulsed lasers. Different from CRDS, CEAS measures the intensity of the transmitted light from the cavity by a photodetector, which reduces the response frequency requirement of the system. Many types of CEAS have been proposed, which can be divided into two categories: integrated cavity output spectroscopy (ICOS) [14,15,16] and mode-locking CEAS. In ICOS, the laser frequency is swept through some cavity modes, and the integrated transmission of these cavity modes is measured. The ICOS-based system is simple in structure, but the cavity transmission is greatly attenuated because of the inefficient injection of light into the cavity, and some noise can be introduced by the non-uniform excitation of the cavity modes [17]. The mode-locking CEAS was reported by Titus Gherman and Daniele Romanini in 2002 [18]. Mode-locked (ML) femtosecond pulsed lasers were used for CEAS in their paper, and the advantages were discussed. After that, mode-locking CEAS was used in a variety of applications by introducing different light sources. In 2008, Michael J. Thorpe et al. coupled the broad spectrum of a mode-locked fiber laser to an optical enhancement cavity to greatly enhance the detection sensitivity of breath samples [19]. In 2014, Chadi Abd Alrahman et al. demonstrated near-infrared cavity-enhanced optical frequency comb spectroscopy of water in a premixed methane/air flat flame. High-power throughput and stable cavity transmission were obtained in their experiment by tightly locking an Er:fiber comb to the cavity that contained the flame [20].

In this paper, a mode-locking CEAS-based system was presented for trace C_2_H_2_ measurement. A continuous-wave distributed feedback (DFB) laser centered at 1530.72 nm was locked to the cavity fundamental mode (TEM_00_ mode) by the Pound–Drever–Hall (PDH) scheme [21] at all times, and the intensity of the cavity-transmitted light was measured continuously to characterize the gas concentration. Since the CEAS-based system did not have an absolute frequency reference, it was difficult to guarantee the long-term stability of the locking frequency. In our system, a new dual-channel feedback-locking loop is proposed to realize laser frequency locking and cavity length locking simultaneously. The drift of the cavity length was suppressed, and the stability of the system was enhanced. The paper is composed as follows. First, selection of the probe absorption line of C_2_H_2_ is reported. Then, the structure of the system and the design of key modules are introduced. Finally, gas detection experiments and the performance of the system are described.

## 2. Absorption Line Selection

The C_2_H_2_ molecule exhibits an overtone rotational–vibrational combination band (v_1_+v_3_ band) around 6550 cm^−1^ as shown in Figure 1. The absorption line located at 6534.36 cm^−1^ (corresponding to 1530.37 nm) was selected as the probe line to detect C_2_H_2_ in this system because of its relatively strong absorption intensity (S = 1.2 × 10^−20^ cm/molecule). Furthermore, the silica fiber has a lower transmission loss in this wavelength, which is suitable for the long-distance transmission of the laser and remote monitoring.

In order to avoid the interference caused by other major gases that exist in the air, the absorbance of the mixed gas of C_2_H_2_ (1 ppm) and air was simulated on the basis of data in the high-resolution transmission molecular absorption database 2012 (HITRAN 2012) [22]. The simulation was conducted in 1 atm, 298 K, and a path length of 100 cm. The absorbance of 1.8 % H_2_O is shown in Figure 2a. The absorption intensity near 6534 cm^−1^ was in 10^−4^ level, which would influence the selectivity of the system. In order to eliminate this interference, a gas purifier (W.A. Hammond Drierite Co., L68GP) was installed before the cavity inlet, able to dry gases to a dryness of 0.005 mg/l, under a flow rate of up to 300 L per hour. The absorbance of other gases present in the air had minimal interference in trace C_2_H_2_ detection, as shown in Figure 2b.

## 3. System Structure and Design of Key Modules

### 3.1. Structure and Configuration

The structure of the cavity-enhanced C_2_H_2_ sensor system is shown in Figure 3. It consists of two modules: the optical module and the electrical module.

(i) Optical module. A DFB laser produced by JDS Uniphase, model number CQF935/908-19570, was used as the optical source. The emission peak wavenumber of the laser is 6534.36 cm^−1^ (1530.72 nm) at an operating temperature of 29 °C and a driving current of 100 mA, which was equal to the peak wavenumber of the selected probe absorption line. The laser is suitable for frequency modulation and locking with a high side-mode suppression ratio and a small linewidth (the linewidth at −3 dB is better than 1 MHz). A fiber isolator (Thorlabs, IO-H-1550FC) was coupled to the laser output polarization-maintaining fiber to attenuate the optical feedback. The isolation of the isolator was 29 dB at a wavelength of ~1.53 μm according to the datasheet provided by the manufacturer. The output fiber of the isolator was coupled to a fiber collimator to convert the fiber-transmitted light into spatially transmitted light and narrow the light beam. The infrared beam passed through a polarizer and was guided into an electro-optic modulator (EOM) for laser frequency modulation. Then, the modulated light was directed to the self-developed cavity after passing through two mode-matching lenses (L5 and L6) and two three-dimensionally adjustable mirrors (M1 and M2). The cavity-transmitted light was focused onto an In-Ga-As detector (PD1, Thorlabs, PDA10CS) by a parabolic mirror (M3), and the cavity-reflected light was directed to another photodetector (PD2, QUBIG, PD-AC200) for laser-cavity frequency locking.

(ii) Electrical module. The electrical part of the system included a commercial laser driver (Vescent photonics, D2-105-500), a three-channel piezoelectric transducer (PZT) controller (Thorlabs, MDT693), and a dual-feedback PDH locking loop. An EOM driver (QUBIG ADU, model 1~200MHz) was used for laser frequency modulation with a modulation frequency of 25 MHz. The cavity-reflected signal from PD2 was mixed with the local oscillator signal in the EOM driver and then filtered to produce the error signal. The error signal was processed by a proportional–integral–derivative (PID) controller 1 (TOPTICA, PID110) to generate a laser current feedback signal. Another PID controller was used in the cavity length-locking loop. A detailed description of the dual-feedback loop is presented in Section 3.4.

### 3.2. Tuning Characteristics of the Laser

The wavelength tuning characteristics of the DFB laser versus the driving current and the operation temperature were measured by a wavelength meter as shown in Figure 4. The tuning index of the temperature was calculated to be 0.14 nm/°C, which was much larger than the tuning index of the driving current (0.0015 nm/mA). Therefore, temperature tuning was adopted to move the wavelength to the desired one, and current tuning was used for PDH locking in this system. An operation temperature of 29 °C combined with a driving current of 100 mA was selected to target the 1530.72 nm C_2_H_2_ absorption line.

### 3.3. Optical ModuleDdesign

The stability condition of the cavity resonant field and the laser cavity mode matching are two main factors that need to be considered in the light path design [23]. For a linear F–P cavity with two concave mirrors, the stability condition is:
(1)$$0<\left(1-\frac{d}{r_{1}}\right)(1-\frac{d}{r_{2}})<1$$
where *r_1_* and *r_2_* are the ROC (radii of curvature) of the cavity mirrors, *d* is the cavity length. In this system, the two cavity mirrors (EKSMA optics) with a same ROC of 100 mm and a calibrated reflectivity of 99.4% at 1.54 μm were spaced by a 15 cm distance. The specifications of the designed cavity were calculated as shown in Table 1.

The mode-matching system was designed to match the spatial distribution of the light beam to the mode distribution of the cavity and obtain a stable and pure fundamental transverse mode in the cavity. In the system, the light passed through a fiber collimator to shape the beam profile and adjust the direction of the exit light. The divergence angle of the fiber collimator varied with the laser wavelength. When the incident laser wavelength was 1530.7 nm, the divergence angle was 0.1294°, and the beam waist radius of the collimated light was calculated to be 219 μm. A concave lens (L5) and a convex lens (L6) were added to the optical path to adjust the beam waist radius. The focal lengths of the two lenses and the lens spacing distances were calculated to generate the correct beam parameters and leave sufficient space for installing other optical components. Two three-dimensionally adjustable mirrors (M1 and M2) were used to adjust the direction and position of the incident laser.

### 3.4. Electro-Optic Modulation-Based PDH Locking Scheme

The dual-feedback locking loop was designed to consist of a PDH locking loop and a cavity length-locking loop, as shown in Figure 5. The electro-optic modulation-based PDH locking loop consisted of an EOM module, a PID controller (PID1), and a laser current control module, which realized the active feedback control of the laser frequency. A 25 MHz sinusoidal signal with a small modulation index was generated by the advanced drive unit (ADU) to modulate the laser frequency. The modulated light entered the cavity and experienced multiple reflections. After that, a portion of light returned from the cavity along the incident path and was collected by the photodetector PD2. This reflected light signal was filtered by a high-pass filter with a cutoff frequency of 200 MHz and multiplied by the phase-shifted reference signal in the EOM driver. Then, an error signal was generated after a low-pass filter (cutoff frequency is 1.9 MHz), which could represent the phase difference between the laser frequency and the cavity resonance frequency. The error signal was then processed by the PID controller (PID1), and a control signal was generated and fed back to the laser current driver for the mode locking between laser and cavity.

To observe the cavity resonance, a periodic sawtooth wave signal was applied to the PZT driver to sweep the cavity length. The measured cavity transmission signal and PDH error signal were captured by an oscilloscope (Tektronix, MDO4104B-6), as shown in Figure 6a. The green line represents the PZT sweep signal, the red line represents the error signal, and the blue line represents the cavity-transmitted signal. The error signal is centered symmetrically and contains a steep linear region near resonance. Figure 6b shows the cavity transmission facula captured by an infrared imager (Electrophysics, PV320). The laser was resonantly coupled to the fundamental mode, and all other high modes in the cavity were eliminated.

### 3.5. Cavity Length Feedback Locking

The optical cavity was used as the reference frequency standard in this system, and the stability of the cavity length determined the performance of the system. During the operation of the system, the cavity experienced a slow frequency drift on cavity length due to variations in the environment (such as temperature, pressure) [24]. This drift caused the deviation of the locked laser frequency from the center of the probe absorption line. In addition, the effective optical path was also changed, causing measurement errors. As shown in Figure 7a, when the laser was locked to the cavity, the transmitted signal was stable, and the mean value of the error signal voltage was close to zero. Without cavity length locking, after 10 min of operation, the fluctuation of the transmitted signal was 15.3 times larger, and the mean value of the error signal changed to −7.7 mV due to the incomplete locking caused by cavity length drift, as shown in Figure 7b.

In order to eliminate the offset of the error signal, a cavity length feedback locking loop was designed, as shown in Figure 5. The controlled object of this feedback loop was the distance between the two cavity mirrors. A PZT was mounted between the incident mirror and the mount with a maximum length variation of 2.5 μm. The low frequency drift of the error signal was extracted by a low-pass filter (low-pass filter 2) with a cutoff frequency of 1 Hz and directed to a PID controller (PID2) to generate a feedback control signal to the PZT driver. The cavity length was then adjusted to suppress the drift.

## 4. Experiment and Results

In order to achieve the environmental stabilization of the system, some measures were taken to reduce mechanical and acoustic vibrations, pressure variations, and temperature fluctuations. The experiment platform was established in a 23 °C thermostatically controlled laboratory. The optical components of the system were fixed to a pressurized air-floating optical table to reduce mechanical noise. The acoustic isolation was provided by a layer of fiberglass insulation wrapped around the cavity. The gas pressure inside the cavity was controlled to be 1 atm during the gas detection measurement by the pressure control module, which consisted of an air pump (KNF lab, model N813.5) and a pressure controller (MKS, type 640). C_2_H_2_ samples with different concentrations were generated by a gas dilution system (Environics, Model 4040), which could produce gas concentrations from percent to ppb levels for single- or multi-point calibration with an accuracy of ±1.0%. All experiments were performed at night to make sure the vibration and acoustic noise levels in the lab were at a minimum.

### 4.1. C_2_H_2_ Absorbance Measurements

In the experiment, the C_2_H_2_ sample with a concentration of 10 ppm balanced with pure N_2_ was pumped into the cavity. The pressure and temperature of the gas were 700 Torr (0.92 atm) and 23 °C, respectively. In order to make sure the laser wavelength swept across the selected C_2_H_2_ absorption line near 6534.36 cm^−1^, the DFB laser current was set to 100 mA, corresponding to a laser power of 18 mW, and the laser operation temperature was modulated from 33.5 °C to 33.7 °C in steps of 0.012 °C, which resulted in the emission peak wavenumber changed from 6533.78 cm^−1^ to 6534.93 cm^−1^ in steps of 0.084 cm^−1^. The laser cavity frequency locking at each wavenumber included two steps. Firstly, the cavity length was adjusted by the PZT driver to make the laser frequency and the cavity resonant frequency roughly the same. Then, the laser frequency was locked to the cavity resonant frequency by a negative feedback current adjustment. The sampling rate of the system was set to 10 Hz, which resulted in 10 data points per second. The average cavity transmission signal voltage within 10 s of each wave number acquired is shown in Figure 8. When the locking wavenumber was equal to 6534.36 cm^−1^, the absorption was in the strongest position, corresponding to an operation temperature of 29 °C.

### 4.2. Calibration and Stability

The cavity-transmitted signal voltage was measured at different C_2_H_2_ concentrations (2 ppm, 4 ppm, 6 ppm, 8 ppm, 10 ppm). At each concentration level, the voltage was recorded for 5 min. The relation between C_2_H_2_ concentration and averaged transmitted signal voltage was acquired as depicted in Figure 9 and confirmed a linear response of the system (R^2^ = 0.997).

The linear fitting equation is:
(2)V(V)=2.44−0.12×C (ppm)


In order to avoid errors caused by the gas distribution system, the stability of the system was investigated by a long-term measurement of pure N_2_. The measurement lasted for 2 h with a sampling rate of 10 Hz, providing 36000 sampling points, calculated as shown in Figure 10a. The system worked steadily and stably during the measurement process without major fluctuations. The limit of detection (LoD) of the system was characterized by Allan deviation analysis as shown in Figure 10b. A LoD of 137 ppb was obtained at an averaging time of 0.1 s which could be further improved to 9 ppb by increasing the averaging time to 11 s. The LOD was valid in a dry gas, free from water.

## 5. Comparison and Conclusion

The comparison among this system and other reported C_2_H_2_ detection systems is shown in Table 2. It was found that the LoD of this system was 9 ppb, which is competitive compared with other reported systems using near-infrared laser sources. Compared to other systems, this sensor system reached a longer effective optical path length (49.7 m) by a compact cavity with a simple structure and small volume. Furthermore, the effective path length could be easily improved by increasing the reflectivity of the cavity mirrors without increasing the technical complexity. Compared to other mode-locking CEAS based systems, a novel cavity length feedback locking loop was introduced in this system to reduce the cavity length drift, which led to good experimental results.

In conclusion, we have developed and characterized a frequency-locked CEAS technique-based C_2_H_2_ sensor system in the near-infrared region. An effective optical path length of 49.7 m was obtained by a 15 cm long cavity. We proposed a dual-feedback locking loop to realize laser frequency locking and cavity length locking simultaneously, which reduced the drift of the cavity length and enhanced measurement stability. A series of C_2_H_2_ detection experiments were carried out, and the results showed the system could work stably for more than 2 h without major fluctuations. The limit of detection of this system reached 137 ppb at an averaging time of 0.1 s, which could be further improved to 9 ppb by increasing the averaging time to 11 s.

## Figures and Tables

**Figure 1 sensors-19-00508-f001:**
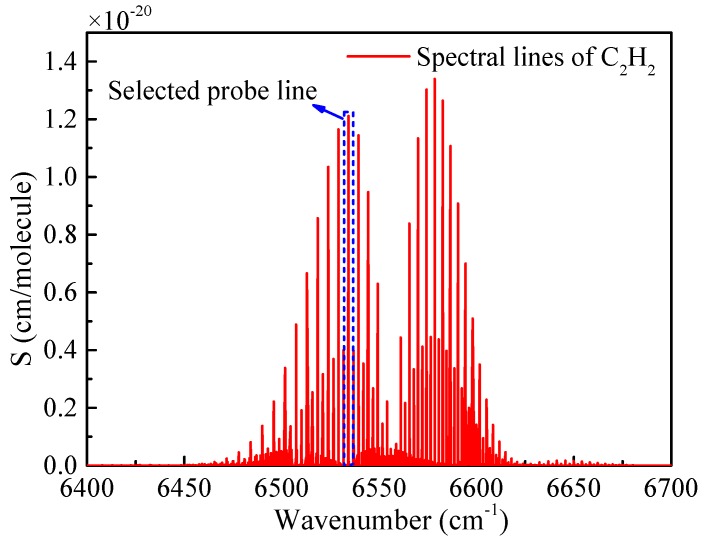
Spectral lines of acetylene (C_2_H_2_) for a spectral range from 6400 cm^−1^ to 6700 cm^−1^.

**Figure 2 sensors-19-00508-f002:**
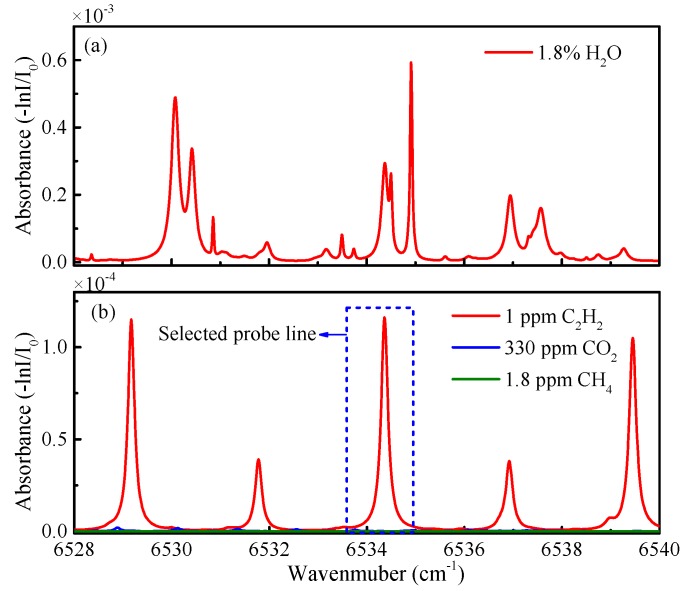
The absorbance of mixed gases from 6528 cm^−1^ to 6540 cm^−1^ at a simulation condition of 1 atm, 298 K, and a path length of 100 cm. (**a**) 1.8% H_2_O (**b**) 1 ppm C_2_H_2_, 330 ppm CO_2_, and 1.8 ppm CH_4_.

**Figure 3 sensors-19-00508-f003:**
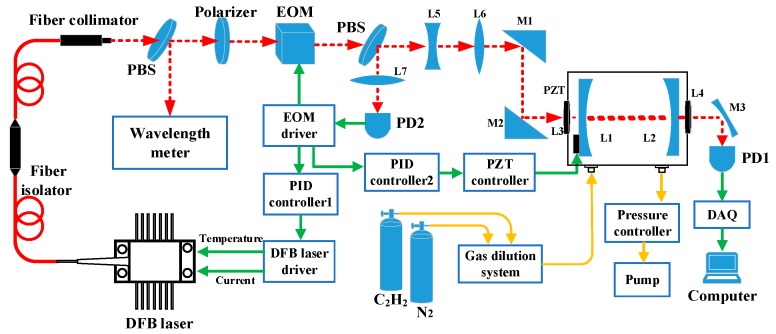
Structure of the frequency-locked cavity-enhanced absorption spectroscopy (CEAS)-based C_2_H_2_ detection system. The red lines represent optical pathways, the green lines represent electrical signal pathways, and the yellow lines are gas flow pathways. PD, photodiode; PBS, polarization beam splitter; DFB, distributed feedback; EOM, electro-optic modulator; PID, proportional–integral–derivative; PZT, piezoelectric transducer; DAQ, data acquisition; L1–7, lens 1-7; M1–3, mirror 1-3.

**Figure 4 sensors-19-00508-f004:**
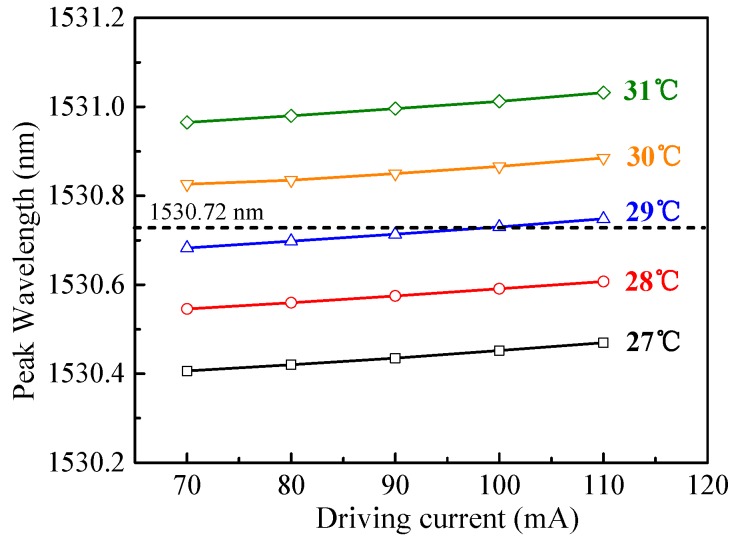
Tuning characteristics of the DFB diode laser.

**Figure 5 sensors-19-00508-f005:**
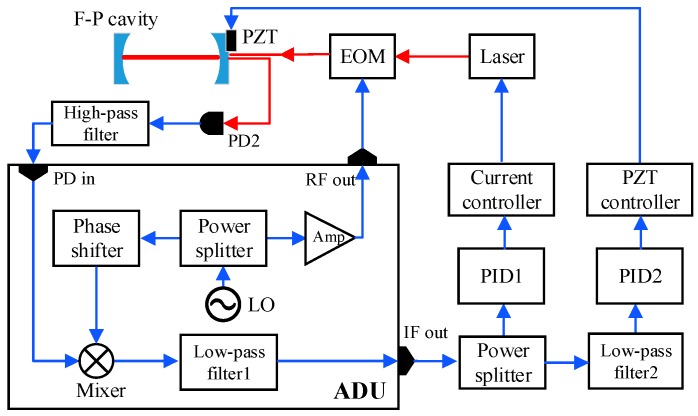
The scheme of the dual-feedback locking loop. LO, Local oscillator; ADU, advanced drive unit; F-P, Fabry–Perot; RF, radio frequency; IF, intermediate frequency.

**Figure 6 sensors-19-00508-f006:**
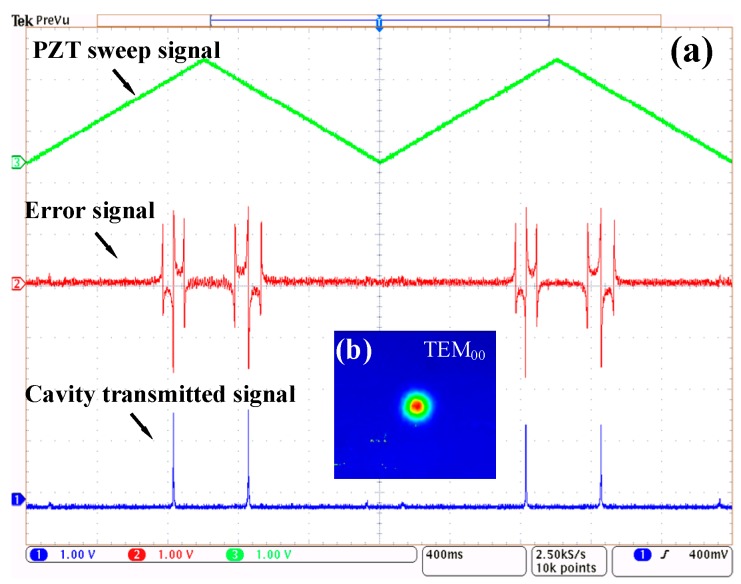
Cavity resonance under critical coupling conditions. (**a**) The green line represents the PZT sweep signal, the red line represents the error signal, and the blue line represents the cavity-transmitted signal. (**b**) Cavity transmission facula.

**Figure 7 sensors-19-00508-f007:**
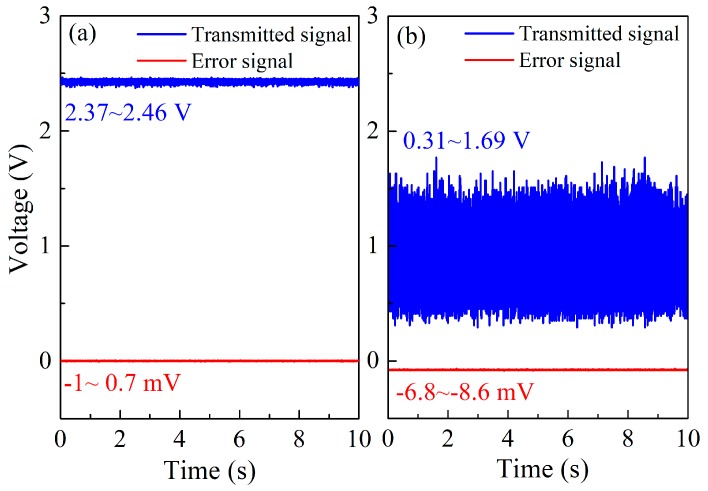
Cavity-transmitted signal and error signal under different conditions. (**a**) The laser is locked to the cavity; (**b**) the laser and cavity are unlocked.

**Figure 8 sensors-19-00508-f008:**
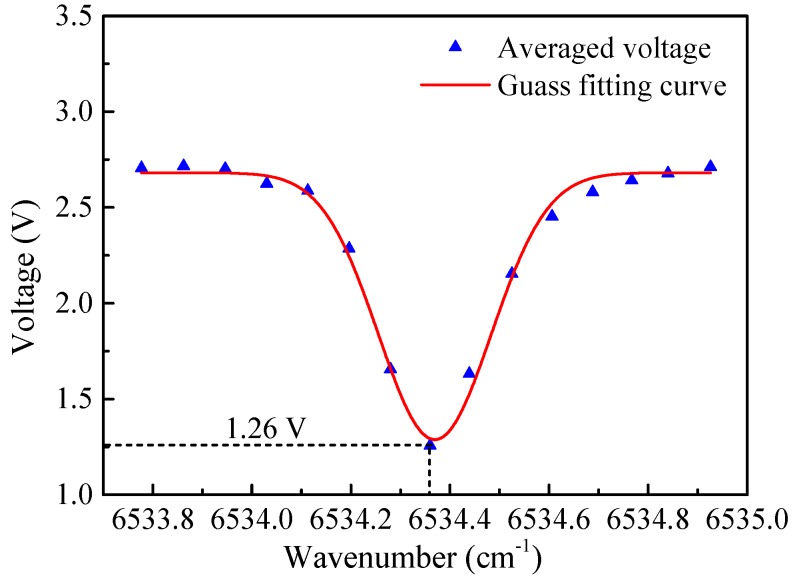
Absorption spectrum of 10 ppm C_2_H_2_.

**Figure 9 sensors-19-00508-f009:**
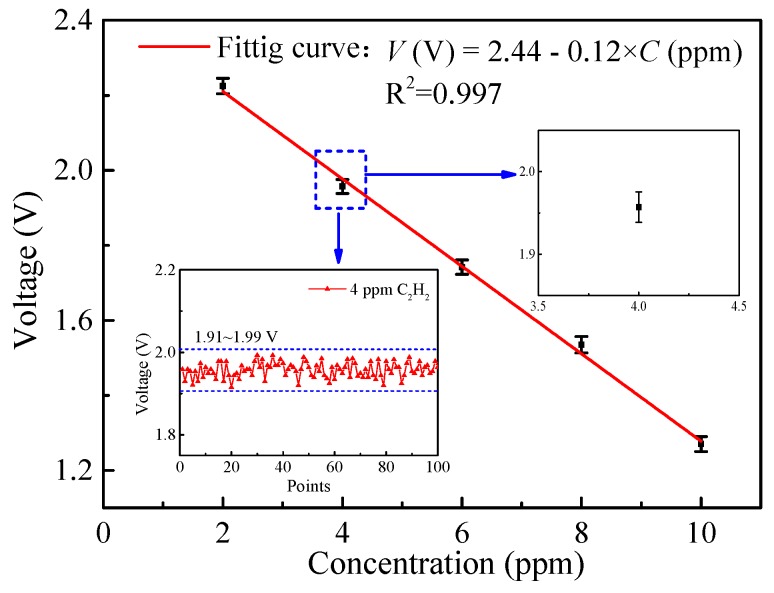
Fitting curve of the cavity-transmitted signal voltage versus C_2_H_2_ concentration. The lower inset shows the measurement results of 4 ppm C_2_H_2_, and the upper inset shows the error bar of the results.

**Figure 10 sensors-19-00508-f010:**
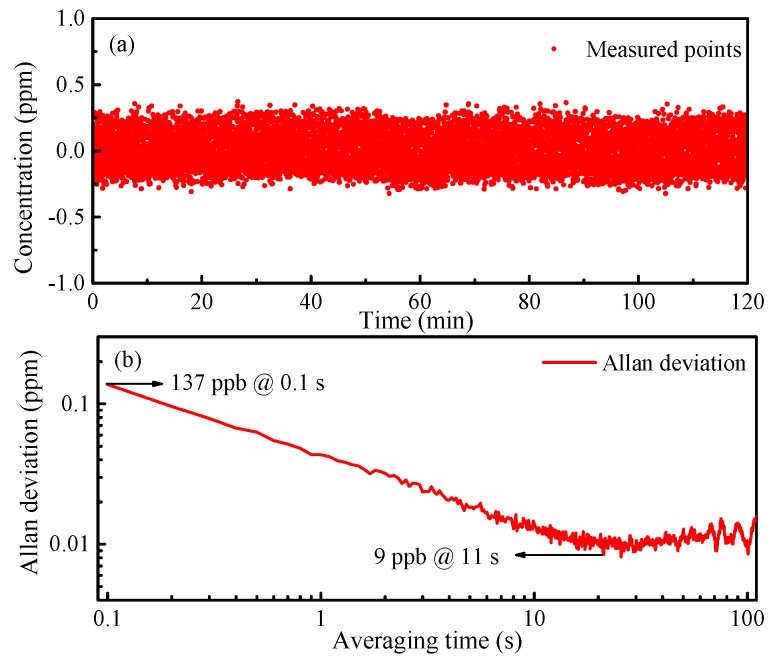
(**a**) Measured concentration of a 0 ppm C_2_H_2_ sample; (**b**) Allan deviation analysis of the sensor system.

**Table 1 sensors-19-00508-t001:** Basic specifications of the designed cavity. FSR, free spectral range.

	Symbol	Value
Cavity length	*L*	15 cm
Reflectivity	*R*	99.4%
FSR	*f_FSR_*	1 GHz
Linewidth	*Δ* *v_c_*	1.92 MHz
Finesse	*f*	520.8
Equivalent Absorption length	*L_eff_*	49.7 m

**Table 2 sensors-19-00508-t002:** Comparison between this system and other reported C_2_H_2_ detection systems. LoD, limit of detection; ICL, interband cascade lasers.

Refs.	Principal	Source Type	Optical Path	LoD
[2]	QEPAS	DFB (1.53 μm)	/	2 ppm
[3]	TDLAS	DFB (1.53 μm)	30 cm	540 ppb
[25]	TDLAS	ICL (3.026 μm)	20.4 cm	1 ppb
[26]	TDLAS	DFB (1.53 μm)	10.1 m	0.49 ppm
[27]	OA-ICOS	DFB (1.53 μm)	9.28 m	85 ppb
This paper	CEAS	DFB (1.53 μm)	49.7 m	9 ppb

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
