# Peer review of "Long-Term Stable Online Acetylene Detection by a CEAS System with Suppression of Cavity Length Drift"

_sensors, 2019, doi:10.3390/s19030508_

Round 1

Reviewer 1 Report

The authors have done a good job presenting their methods and results.  They need to include more of the prior work specifically in mode-locking CEAS, and need to distinguish their approach from this past work.  

Author Response

Dear Reviewer #1,

We appreciate your helpful suggestions for improving our manuscript, which were included in our revised manuscript ID 419587. Based on Reviewer #1’s advices, we revised our MS # 419587.  

More information about the prior work in mode-locking CEAS and the comparison among this system and other reported C2H2 detection systems were added in the revised manuscript. 

Please refer to the red sentences in Section 1 on Page 2 and the underlined red sentences in Section 5 on Page 9.

Yours sincerely,

Qixin He

Reviewer 2 Report

This is a good paper on acetylene detection using mode-locked CEAS. Following are my comments:

1) How much did the gas purifier remove water vapor? Provide these data.

2) You discuss interference issues in early figures but later measurements are all done without any interfering species present. This is a weak aspect of the paper. 

3) How did you determine the laser linewdith?

4) Why did you use a modulation frequency of 25 MHz for the EOM? How was this selected?

5) Why relatively low reflectivity mirrors were used? Could you do similar work with higher R mirrors?

6) Did you verify mirror R or you just replied on manufacturer specs? This could lead to big errors.

7) What was laser power?

8) Figure 7 caption should be more clear to say (a) is with locking and (b) is without. 

9) Give more details of your gas dilution system and its accuracy / uncertainty. 

10) Figure 8: 10 ppm of C2H2 in what bath gas? What was pressure and temperature?

11) Figure 8: How can you decrease the spectral separation between the points? 

12) Why sampling rate of 10 Hz was selected?

13) You should compare your obtained detection limit with other C2H2 works in literature. 

Author Response

Dear Reviewer #2, 

We appreciate your helpful suggestions for improving our manuscript, which were included in our revised manuscript ID 419587. Based on Reviewer #2’s suggestions, we revised our MS # 419587.  

A detailed response as well as reviewer comments can be found in the attatched word file.

Yours sincerely,

 Qixin He
